# Fire suppression makes wildfires more severe and accentuates impacts of climate change and fuel accumulation

Mark R. Kreider [1] ✉, Philip E. Higuera[2], Sean A. Parks [3], William L. Rice[4], Nadia White[5] & Andrew J. Larson [1,6]

Fire suppression is the primary management response to wildfires in many areas globally. By removing less-extreme wildfires, this approach ensures that remaining wildfires burn under more extreme conditions. Here, we term this the "suppression bias" and use a simulation model to highlight how this bias fundamentally impacts wildfire activity, independent of fuel accumulation and climate change. We illustrate how attempting to suppress all wildfires necessarily means that fires will burn with more severe and less diverse ecological impacts, with burned area increasing at faster rates than expected from fuel accumulation or climate change. Over a human lifespan, the modeled impacts of the suppression bias exceed those from fuel accumulation or climate change alone, suggesting that suppression may exert a significant and underappreciated influence on patterns of fire globally. Managing wildfires to safely burn under low and moderate conditions is thus a critical tool to address the growing wildfire crisis.

Wildfires are becoming more destructive and deadly around the world[1–4]. The societal and ecological impacts of fires[5,6] are in our collective consciousness—from Australia's 2019–2020 megafires[7], to destructive wildfires in the Mediterranean[8], to beloved giant sequoias killed by fire in California[9]—prompting widespread calls to address the wildfire crisis[8,10–14]. We understand the broad drivers of increasing fire activity: changes in climate[1,15,16], vegetation and fuel accumulation[8,17,18], and ignition patterns[19]. However, humans also play a direct role in modifying fire activity across much of the globe by engaging with fires minutes to hours after ignition (i.e., initial attack[20,21]), and subsequent suppression of escaped fires[22–24]. While weather, fuels, topography, and ignitions determine how fires might burn, humans strongly shape this into when, where, and how fires do burn (Fig. 1).

Wildfires only burn if they are not extinguished through suppression. Thus, suppression is a "filter" that allows certain types of fire to pass through while removing other types of fire (Fig. 1b). In some

locations (e.g., a remote wilderness area) this filter may be relatively porous, and many fires may burn with only minimal suppression[25]. In most landscapes, however, aggressive suppression of fire is a cultural expectation, and the suppression filter is much less permeable, with only the most extreme fires escaping (e.g., Maximum suppression in Fig. 1)[20,21]. Intentional fires (i.e., prescribed fires and cultural burning) are the only types of fires that do not pass through the suppression filter, as they are allowed to burn unimpeded if they are within prescription (Fig. 1a). Area that does not burn because it was "removed" through suppression, results in fuel accumulation and ultimately increases the likelihood and intensity of future fires[26,27] (Fig. 1a). This well-known consequence has been termed the "fire suppression paradox"[28,29]: by putting out a fire today, we make fires harder to put out in the future[26,30–32] (Table 1).

Suppressing wildfires also has an additional and poorly quantified consequence that we define as the "suppression bias":

[1]Department of Forest Management, University of Montana, Missoula, MT 59812, USA. [2]Department of Ecosystem and Conservation Sciences, University of Montana, Missoula, MT 59812, USA. [3]Aldo Leopold Wilderness Research Institute, Rocky Mountain Research Station, USDA Forest Service, Missoula, MT 59801, USA. [4]Department of Society and Conservation, University of Montana, Missoula, MT 59812, USA. [5]Environmental Science and Natural Resource Journalism, University of Montana, Missoula, MT 59812, USA. [6]Wilderness Institute, University of Montana, Missoula, MT 59812, USA. ✉e-mail: mark.kreider@umontana.edu

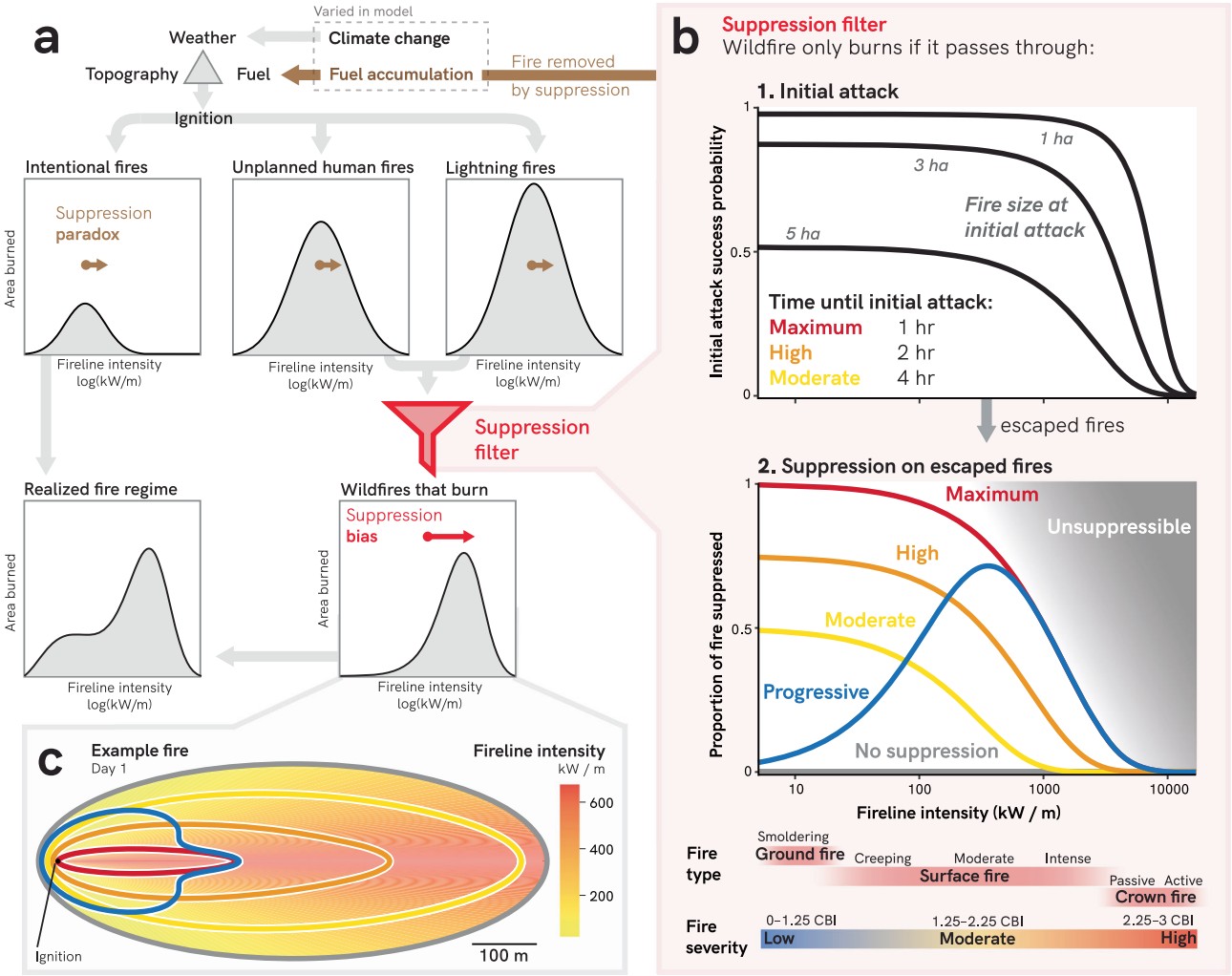

**Fig. 1 | Conceptual diagram of how suppression influences fire. a** Potential fire behavior depends on the fire triangle (topography, weather, fuel) and ignitions. Intentional ignitions (i.e., prescribed fires and cultural burning) do not pass through the suppression filter, as they are allowed to burn unimpeded if within prescription. Unplanned human ignitions and lightning ignitions only burn if they successfully pass through the "suppression filter." Fire "removed" by the suppression filter leads to fuel accumulation, influencing fires from all ignition types (suppression paradox, brown color). Wildfires that do burn are biased toward the fire that was not removed (suppression bias, red). These wildfires, together with intentional fires, form the realized fire regime with the suppression paradox and suppression bias inherently incorporated. **b** The suppression filter. 1) Initial attack success probability as a function of fireline intensity and fire size at initial attack (from Hirsch et al.[69]); 2) Proportion of escaped fire suppressed as a function of fire intensity. Suppression becomes increasingly impossible at high fire intensities. Colors depict the suppression scenarios used in the simulation. **c** Fire perimeters (viewed from overhead) after the first day of burning for an example ignition. Colors correspond to suppression scenarios shown in panel **b**. Fire intensity of the burned area is displayed with a color ramp.

### Table 1 | Comparison of the fire suppression paradox and bias

| Term | Definition | Mechanism | Impact | Time lag |
|---|---|---|---|---|
| Fire suppression paradox | By suppressing fire today, we increase fuel loads, making fires harder to suppress in the future | Fuel accumulation | Indirect | Future |
| Fire suppression bias | By suppressing some fire types more than others, the remainder reflects a biased representation of fire types | Differential suppression filter | Direct | Immediate |

fire suppression extinguishes some types of fire (e.g., surface fire) more than others (e.g., crown fire), and thus skews the resulting fire activity toward those types less likely to be removed (Table 1). In contemporary fire management, which easily suppresses and removes low-intensity fire, this bias is inevitably toward higher-intensity burning occurring under extreme weather[20,21,33–35]. Thus, the fires which ecosystems, species, and people experience are skewed towards the most severe and destructive.

We define management approaches that suppress lower-intensity fire more heavily than higher-intensity fire as "regressive suppression," borrowing language from economics (e.g., a regressive tax rate decreases as taxable income increases) (Table 2). In some instances, however, management could contain and suppress relatively higher-intensity fire more heavily than lower-intensity fire, an approach we term "progressive suppression" (e.g., a progressive tax rate increases as taxable income increases) (Table 2). Both regressive and progressive suppression are subject to the same upper limit, above which fires are

**Table 2 | How the type of suppression influences the resulting suppression bias**

| Type of suppression | Definition | Direction of suppression bias |
|---|---|---|
| Regressive suppression | Suppresses lower-intensity fire more heavily than higher-intensity fire | Toward higher-intensity fires, more extreme fires |
| Progressive suppression | Suppresses higher-intensity fire more heavily than lower-intensity fire | Toward lower-intensity fires, more moderate fires |

simply too intense to suppress (Fig. 1b)[36]; however, within the domain where suppression is possible, regressive and progressive approaches can have profoundly different impacts (i.e., biases) on the way fires burn.

Although the fire suppression bias has been referenced tangentially in the literature[8,28,33,37–39], the emergent impacts of the suppression bias have not been assessed. This is largely due to the difficulty of isolating the impact of suppression with empirical data. Suppression is so ubiquitous that we have virtually no control landscapes where fire is completely unsuppressed; even in remote wilderness areas, some fires are still suppressed[40]. Furthermore, it is difficult to measure the magnitude of suppression efforts because even relatively direct proxies such as suppression cost are confounded by other factors, including terrain accessibility, human infrastructure at risk, and availability of suppression resources[41]. Finally, data on suppression efforts are generally only available for larger fires[42], obscuring the many ignitions that are quickly and easily suppressed during initial attack[20,21]. To overcome these constraints, we used a simulation approach to assess and quantify the magnitude of the fire suppression bias on fire behavior and ecological impacts, relative to the influence of climate change and fuel accumulation.

Our modeling framework simulates fundamental components of fires: weather and fuel moisture; ignitions; fire growth; fire suppression (through initial attack and containment of escaped fires); and ecological effects. To isolate the effect of fire suppression, we simulated thousands of fires with identical biophysical conditions, but which differed only in their suppression scenario, including three "regressive suppression" scenarios (Moderate, High, and Maximum; Fig. 1b), one "progressive suppression" scenario (Progressive; Fig. 1b), and a control scenario with no suppression. For each fire, we calculated the proportion burned at high severity, average fire severity, daily and total fire size, and the diversity of fire severity[43]. To compare the influence of suppression to that of climate change and fuel accumulation, we simulated fires across a range of plausible current and future fuel aridity (vapor pressure deficit; VPD) and fuel loading conditions in forest ecosystems in North America. These ranges represent a 240-year time period of modeled increases (e.g., increased VPD based on RCP 8.5 climate scenario[44]; fuel loading rates based on historical fuel modeling[45]).

Using this modeling framework, we show how the suppression bias directly influences fire activity and subsequent fire effects. Specifically, we asked: 1) How does fire suppression influence patterns of area burned, ecological impacts (i.e., fire severity), and the diversity of both factors over space and time? 2) How does the magnitude of this influence compare to that from climate change and fuel accumulation?

## Results

### Regressive fire suppression makes fires more severe
Across the simulated range of fuel aridity, regressive suppression scenarios (Moderate, High, and Maximum) increased the ecological impacts of wildfire, as reflected by higher fire severity metrics. We simulated fire severity by linking fire intensity to the Composite Burn Index (CBI), which runs from zero (unburned) to three (maximum fire severity), with values above 2.25 considered high-severity[46]. Under the Maximum fire suppression scenario, a greater proportion of each wildfire burned at high severity (Fig. 2a, b). On average, wildfires burning under the Maximum suppression scenario had over twice as high proportion that burned at high severity, compared to fires

burning with no suppression. Across all fuel aridity simulations, Maximum suppression increased mean fire severity by an average of 0.21 CBI units, relative to fires with no suppression (Fig. 2c). This increase in fire severity is equivalent to the cumulative effects of 102 years of increased fuel aridity from climate change alone, under the no-suppression scenario (i.e., an increase in mean summer VPD of +0.85 kPa) (Fig. 2c inset). Similarly, across the simulated range of fuel loading values, Maximum suppression increased mean fire severity by an average of 0.22 CBI units, relative to control scenarios (Fig. 2d). This increase is equivalent to the effect of 102 years of additional fuel accumulation under no fire suppression (i.e., an increase in 100-h surface fuel loading of +3.7 Mg ha$^{-1}$) (Fig. 2d inset).

In contrast, Progressive suppression reduced the proportion of each wildfire that burned at high severity, for most levels of fuel aridity and fuel loading (Fig. 2a, b). Compared to fires burning with no suppression, wildfires burning under the Progressive suppression scenario had an average of 17% (across fuel aridity gradient) and 15% (across fuel loading gradient) less proportion that burned at high severity. Mean fire severity was also lower under the Progressive fire suppression scenario, across most of the simulated range of fuel aridity and loading, with an average reduction of 0.04 and 0.03 CBI units, respectively (Fig. 2c, d). Given modeled rates of fuel aridity and fuel loading change, these differences are equivalent to burning under scenarios of no suppression from 17 and 14 years in the past (i.e., changes of −0.15 kPa VPD and −0.49 Mg ha$^{-1}$), respectively (Fig. 2c, d insets).

### Regressive fire suppression accentuates trends of increasing area burned
While increasing fuel aridity and fuel loading led to a rise in area burned for wildfires under all scenarios (i.e., suppression or not), fires under the regressive suppression scenarios displayed higher sensitivity to increasing fuel aridity and fuel loading (Fig. 3). Across the gradient of increasing fuel aridity, area burned under the Maximum suppression scenario increased by 5.0% per year, compared to only 1.8% per year for wildfires under the no-suppression scenario. Thus, across the 240-year range of increased fuel aridity, yearly burned area doubled nearly three times as fast under Maximum fire suppression, compared to scenarios without fire suppression (i.e., 14 vs. 39 years). This difference was even more marked under increasing fuel loading: area burned under the Maximum suppression scenario increased by 3.7% per year, compared to only 0.7% per year for wildfires that were not suppressed (Fig. 3). Thus, across the 240-year range of increased fuel loading, yearly burned area doubled over five times faster under Maximum suppression, compared to scenarios without fire suppression (i.e., 19 vs. 94 years). Fires simulated under the Progressive suppression scenario had the lowest sensitivity to increasing fuel aridity and fuel loading of any suppression strategy, including no suppression. Yearly burned area doubled every 44 years across the simulated increases in fuel aridity, and only every 133 years across the simulated gradient of fuel accumulation. Patterns were similar for area burned at high severity, with regressive scenarios leading to faster proportional increases in area burned and the Progressive scenario maintaining the slowest increase in area burned (Supplementary Fig. S1).

### Regressive fire suppression decreases the diversity of fire effects
Relative to unsuppressed fires, regressive suppression decreased the diversity of fire effects (i.e., diversity of burn severity) at all levels of fuel aridity and fuel loading values (Fig. 4). In the Maximum

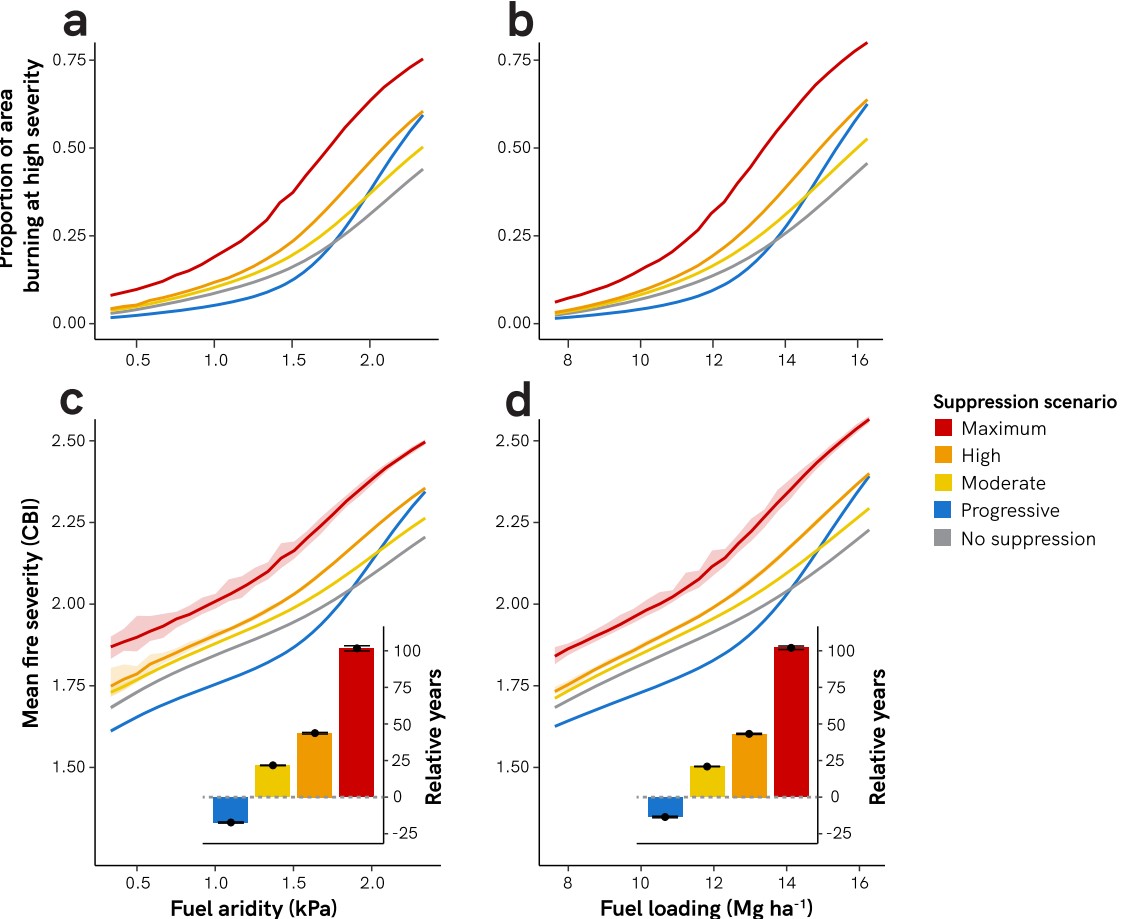

**Fig. 2 | Effects of fire suppression on fire severity.** Panels **a** and **b** show the proportion of high-severity fire (CBI > 2.25) across ranges of fuel aridity and fuel loading. Panels **c** and **d** show mean fire severity across ranges of fuel aridity and fuel loading. Insets show the average number of years of modeled climate change (vapor pressure deficit increase of 0.008 kPa yr$^{-1}$) or fuel accumulation (100-h fuel accumulation rate of 0.036 Mg ha$^{-1}$ yr$^{-1}$) to yield the difference in fire severity between suppressed and unsuppressed fires. Variability across the 40 simulation replications is shown with 95% confidence intervals (too small to see for some) or error bars (insets on **c** and **d**). Fuel loading in panel **b** and **d** depicts 100-h surface fuel loading values. Simulations across the fuel aridity range were run at a constant 100-h surface fuel loading of 11.23 Mg ha$^{-1}$; simulations across the fuel loading range were run at constant mean fire season vapor pressure deficit of 1.17 kPa.

suppression scenario, 97–99% of fires were contained under 121 ha (300 ac) for all but the most extreme levels of fuel aridity and loading (Supplementary Fig. S2). Under the regressive fire suppression scenarios, a higher proportion of area burned came from a small proportion of extreme fires (Fig. 4c). For example, under the Maximum suppression scenario, 91% of the area burned came from the largest 1% of fires, compared to only 4% of the area burned that came from the largest 1% of fires under the scenario of no fire suppression. In contrast, under the Progressive fire suppression scenario, fires had the highest diversity of fire effects across the entire range of fuel aridity and fuel loading values, with especially pronounced increases in diversity under more extreme conditions. Progressive suppression also led to fires with the most even distribution of area burned across fires of any suppression scenario (Fig. 4c).

## Discussion

We show with a simulation experiment how decades of regressive fire suppression have likely contributed to observed rates of increased burned area and high-severity area burned, independent of climate change and fuel accumulation. While some wildfires have always burned under extreme conditions and at high severity, the fire suppression bias magnifies the proportional representation of these fires by removing fires that would have burned with low or moderate severity (Supplementary Fig. S3). The result is akin to the overprescription of antibiotics: in our attempt to eliminate all fires, we have

only eliminated the less intense fires (that may best align with management objectives such as fuel reduction[47]) and instead selected for primarily the most extreme events (suppression bias) and created higher fuel loads and more "suppression-resistant" fires (suppression paradox). Through regressive fire suppression, we are effectively bringing a more severe future to the present—experiencing average fire severities that would not otherwise happen for a century. Our findings suggest that the abnormally high proportions of high-severity fire witnessed in many areas globally (e.g. refs. 48–51), is due, in part, to the influence of the suppression bias itself.

The suppression bias also has profound impacts on society and social perceptions of fire. By disproportionately removing fires with desirable impacts[47], regressive fire suppression ensures that most people interact with wildfires that burn during extreme events. This in turn makes it less likely for individuals to value the beneficial aspects of less-extreme fires, and less likely to support or desire active fire management[52], further exacerbating the suppression bias[10].

Though regressive suppression keeps many fires at small sizes (Supplementary Fig. S2) and reduces the absolute amount of burned area (relative to a world with no suppression), in the face of climate change and fuel accumulation, it counter-intuitively leads to a higher relative rate of increasing area burned over time. This is because regressive suppression amplifies the difference between the amount of fire that burns under less fire-conducive climates (e.g., in the recent past) and how much will burn despite heavy suppression under more

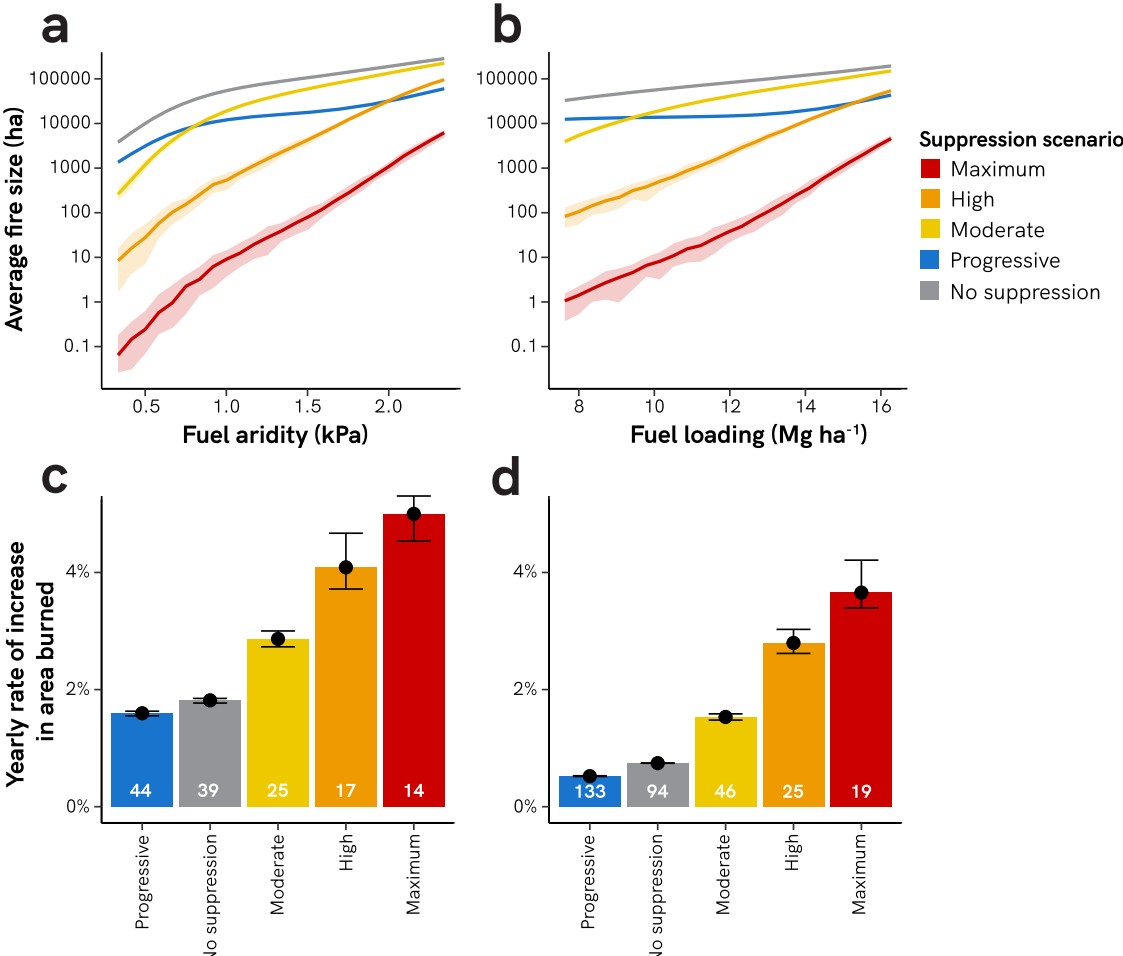

**Fig. 3 | Effects of fire suppression on burned area increase.** Panels **a** and **b** show trends in average fire size across ranges of fuel aridity and fuel loading. Fuel loading in panel **b** depicts 100-h surface fuel loading values. Yearly rates of increase in panels **c** and **d** are calculated with a yearly increase in fuel aridity of 0.008 kPa yr$^{-1}$ or a yearly increase in fuel accumulation (100-h fuel accumulation rate) of 0.036 Mg ha$^{-1}$ yr$^{-1}$, respectively. White numbers at the base of bars are the doubling time, in years, of burned area. Variability across the 40 simulation replications is shown with 95% confidence intervals (**a** and **b**; too small to see on some curves) or error bars (**c** and **d**). Simulations across the fuel aridity range were run at a constant 100-h surface fuel loading of 11.23 Mg ha$^{-1}$; simulations across the fuel loading range were run at constant mean fire season vapor pressure deficit of 1.17 kPa.

fire-conducive future climates. For example, across simulated climate change, area burned doubled nearly twice as quickly under regressive suppression, compared to not suppressing fires at all. Our work thus suggests that observed high rates of increasing area burned around the world (e.g. refs. [3,53]) are at least partially driven by the suppression bias. People and societies are adapted to what they experience to be "normal," and deviations from this baseline require adaptation; by causing the baseline to shift at an even faster rate, regressive suppression further heightens the stress on societies responding to changing conditions[54,55].

The fire suppression bias also has fundamental effects on the longstanding ecological and evolutionary role of fire in terrestrial ecosystems[56]. Fire suppression not only reduces how often plants and animals are exposed to fire, which is detrimental to fire-dependent organisms[57], but it also guarantees that a greater proportion of these encounters are with high-intensity fire. By preferentially removing low-intensity fire through regressive suppression, we have likely shifted the selective pressures of natural selection, unintentionally favoring traits that confer resistance or resilience to high-intensity fire over traits supporting persistence through lower-intensity fire. Fire is also an important catalyst for community reorganization and adaptation in the face of changing environmental conditions such as global

warming[58]. However, by reducing the prevalence of fire, suppression limits opportunities for reorganization; instead, ecosystems accumulate inertia from the current species composition and structure, which may not be well-aligned to future conditions[59–61]. Furthermore, because regressive suppression biases fire toward more severe conditions, with decreased seed and propagule availability and more stressful post-fire climatic environments, any reorganization that does occur is more likely to lead to state-shifts[61–64].

We demonstrate that progressive suppression leads to less extreme simulated fire behavior and effects. Indeed, empirical data from protected areas (which tend to have lower rates of suppression[65] and may represent the closest existing examples to progressive suppression) show lower fire severities[66,67] and a greater diversity of fire effects[43], consistent with our conclusions. In our simulations, fires under the Progressive suppression scenario had equivalent fire severity to unsuppressed fires burning under less fire-conducive conditions—in other words, effectively reversing the impacts of climate change or fuel accumulation by one to nearly two decades. Area burned under the Progressive suppression scenario also doubled much slower in response to climate change, compared to regressive suppression scenarios. A society living under progressive suppression would be less stressed by climate change, as their perceived "normal"

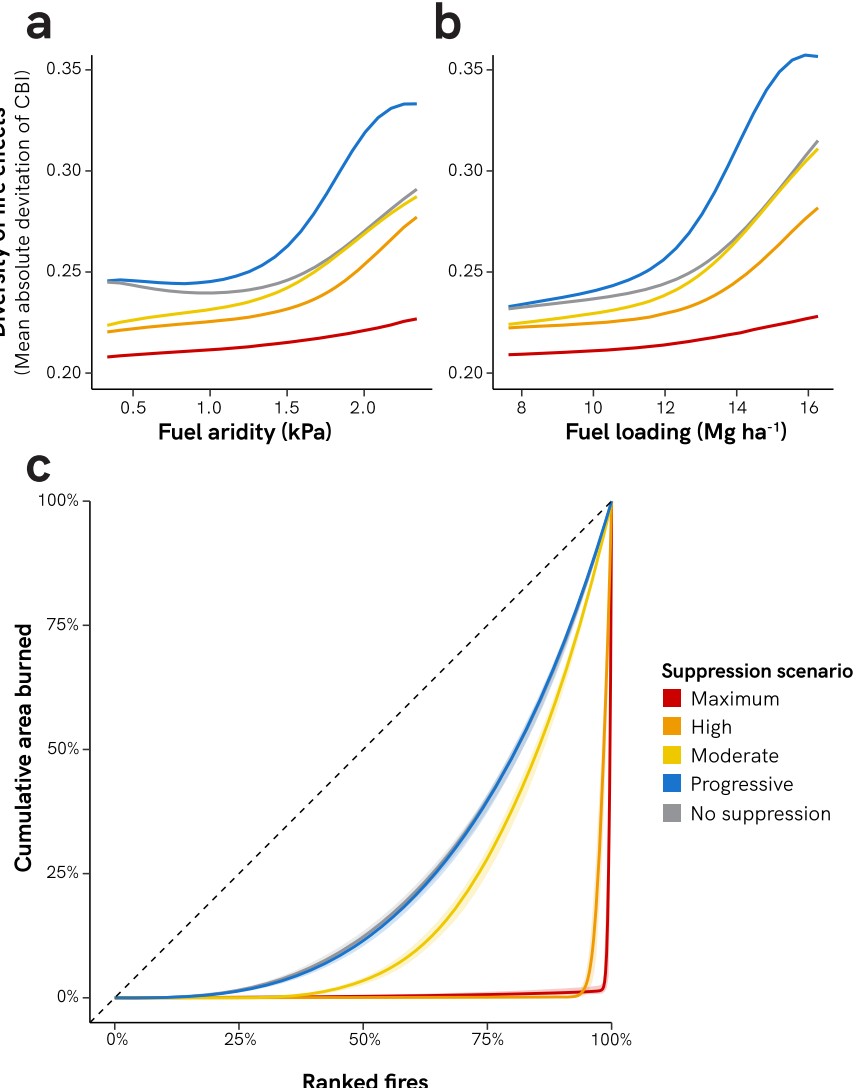

**Fig. 4 | Effects of fire suppression on diversity of fire effects.** Panels **a** and **b** show the effects of fire suppression on the diversity of fire effects across ranges of fuel aridity and fuel loading. Diversity of fire effects is calculated as the mean absolute deviation of fire severity (CBI) sensu Steel and colleagues[43]. Fuel loading in panel **b** depicts 100-h surface fuel loading values. Simulations across the fuel aridity range were run at a constant 100-h surface fuel loading of 11.23 Mg ha$^{-1}$; simulations across the fuel loading range were run at constant mean fire season vapor pressure deficit of 1.17 kPa. **c** Lorenz curves for each suppression scenario; fires are ranked by increasing area burned. Simulations run at mean seasonal vapor pressure deficit of 1.17 kPa and fuel loading of 11.23 Mg ha$^{-1}$ (100-h fuel load). The dashed line represents hypothetical fire activity where the area burned is spread equally across all fire days. Variability across the 40 simulation replications is shown with 95% confidence intervals but which are too small to see for some curves.

conditions would change half as fast. By allowing more lower-intensity fire, progressive suppression could buy time, helping societies and ecosystems adapt to climate change[54].

Our simulations are grounded in fundamental physical aspects of fire behavior[68] and reveal important and underappreciated consequences of fire suppression; however, our model is not intended to predict fine-scale fire behavior. For example, we did not incorporate spatial variability in topography, wind direction, or fuel loading within any individual fire. Additionally, each modeled fire is unaffected by fires that have occurred in prior years (i.e., the model does not incorporate the effects of the fire suppression paradox; Table 1). This may make our results a conservative estimate of the total impact of regressive suppression, since landscapes under regressive suppression would accumulate fuel faster than those with no suppression, further heightening the difference in fire intensity.

While our model includes practical implementations of suppression[36,69], it does not incorporate dynamic resource allocation

as the number of fires increase across a landscape. This means it does not explicitly account for scenarios where suppression resources are depleted from numerous fires burning simultaneously, such as during National Wildland Fire Preparedness Level 5 in the U.S.[41,70]. Consequently, the simulation may overestimate suppression effectiveness and underestimate suppressed fire sizes in these conditions. However, the model assumes near-complete ineffectiveness of suppression during extreme weather events when fire intensity is high (Fig. 1b), effectively incorporating resource scarcity. Regardless, in conditions when fire suppression is effective, our results show that it inherently biases resulting fire patterns.

Our model generates patterns of fire behavior that align with empirical data. For example, burned area patterns under Maximum suppression (e.g., nearly all fires contained before reaching 121 ha (300 ac); 1% of fires accounting for 91% of area burned) closely resemble long-term trends observed in the U.S. (97–99% of fires contained before reaching 121 ha (300 ac); 1% of fires accounting for 98% of area

burned[71]). Thus, our results reveal important general expectations of the impacts of fire suppression, relevant to any flammable location worldwide where suppression is used. Future empirical work can test these expected impacts across a variety of ecological and cultural settings.

Although safely allowing low- and moderate-intensity fire is essential for learning to co-exist with wildfire[72], implementing progressive suppression faces a range of challenges that span social-ecological systems[41]. Numerous land management agencies have ingrained cultures and policies that incentivize regressive fire suppression[8,10,33], and shifts in management may be hampered by a lack of trust and support from the public[73], in addition to the widespread public expectation that all wildfires should be suppressed[74]. Progressive suppression would let low-intensity fire spread relatively unencumbered, while more strongly suppressing higher-intensity fire —a management approach that may not always be practical, safe, or possible. Additionally, the operating space for progressive suppression (i.e., maintaining a gradient where higher-intensity fire is more heavily suppressed than lower-intensity fire) is increasingly constrained as conditions become more fire-conducive (Supplementary Fig. S4), and as more valued human resources are built in flammable environments[5]. Finally, while lower- and moderate-intensity fires can reduce the smoke-related health impacts of large, high-intensity wildfires[75], moving toward progressive suppression would likely increase the frequency of low-level smoke emissions, much like the increased use of prescribed fire[75,76]. As such, paradigm shifts in fire management would necessitate public health interventions at all levels: from individual behavioral strategies to public policies and community resources (e.g., facilitating the use of publicly available air-quality data, subsidizing or providing high efficiency particulate air [HEPA] air filters, and creating publicly accessible clean-air spaces)[4,77].

Even when and where progressive suppression is infeasible, our results show that less aggressive implementations of regressive suppression (e.g., moving from Maximum to Moderate suppression, analogous to calls for increased fire use[10]) can dramatically reduce the suppression bias. Adaptive management frameworks that facilitate risk-informed differences in management approaches (e.g., the PODs framework[78]) could likewise help lessen the impact of the suppression bias. For example, such fire management could implement regressive suppression approaches when necessary (e.g., near human infrastructure) and progressive or no suppression approaches when and where more feasible. Finally, intentional fires (i.e., prescribed fires and cultural burning; Fig. 1) also play an important role in tandem with progressive suppression[10,79,80]. While these practices support a range of values on their own[47], they also facilitate the implementation of progressive suppression strategies by introducing low-intensity fire and creating landscape heterogeneity.

We demonstrate that the suppression bias is a major driver of fire activity and ecological impacts. While the negative impacts of fuel accumulation are commonly recognized as an indirect consequence of fire suppression, integrating the impacts of the suppression bias would improve our understanding of fire-human relationships, and ongoing changes in fire activity. Part of the solution to coexisting with wildfire now and into the future requires developing and applying technologies and approaches that allow us to safely manage wildfires under moderate burning conditions. Arguably, this will be as effective as other needed interventions, such as mitigating global warming, minimizing unintentional human-related ignitions, and modifying forest structure to reduce fire severity when fires occur.

## Methods
Our model simulates individual fire events independently, and it is not intended to represent a specific real-world landscape. All models are tradeoffs between fine-scale precision and large-scale

generalizability[81], and we designed our simulation approach to be generalizable while incorporating fundamental physical aspects of fire behavior. We use a wide range of fuel loading and climate parameters, to both demonstrate the global applicability of our results and address the variability and uncertainty inherent in complex real-world environments[82] and demonstrate that the consequences of the suppression bias are not unique to a specific biophysical setting (i.e., a fire regime). The parameters and simulations described below are generally informed by variables and ranges of variability representing forest ecosystems in western North America. Full details are provided in the Supplementary Information (Supplementary Methods).

### Simulating ignitions & fire spread
For each fire, we randomly chose an ignition day from a hypothetical 150-day fire season. Ignitions could smolder for up to three days, during which fire spread would occur if the daily fuel moisture was less than the moisture of extinction (25%). If fuel moisture never fell below 25% during this three-day period, the ignition was assumed to be extinguished naturally.

We assumed elliptical fire shape[83] and modeled daily fire growth based on Huygens' principle[84], which assumes that the growth of each point on the fire perimeter can be independently modeled as an expanding ellipse. The shape of this ellipse becomes longer and narrower at higher wind speeds[83] (Supplementary Fig. S5). We modeled daily heading-direction fire spread rate, using the rothermel function in the R package firebehavioR[85], which incorporates potential transitions to crown fire spread[86,87] (Supplementary Fig. S5). We converted the modeled heading-direction fire spread rates to elliptical expansion factors and distances at any angle from the ignition (Supplementary Methods). We calculated two metrics of fire intensity—fireline intensity and flame length—using equations from the rothermel function in the R package firebehavioR[85] and extended them to elliptical fireline intensity using equations of Catchpole and colleagues[88]. We estimated fire severity by linking flame length to estimated tree mortality from mixed-conifer ecosystems[89] and finally to Composite Burn Index (CBI), a measure of fire-caused vegetation mortality and soil organic matter consumption[46].

### Simulating suppression
We tested several different suppression scenarios: three regressive suppression scenarios (Moderate, High, Maximum), one progressive suppression scenario (Progressive), and one control scenario (No suppression). We simulated suppression of fires with a two-step process: 1) initial attack, and 2) subsequent suppression of fires that escape initial attack[23,24]. To simulate initial attack success, we used a modeled relationship from Hirsch and colleagues[69] that estimates the probability of escape as a function of heading fireline intensity and fire size at the time of initial engagement. For regressive suppression scenarios, we stochastically simulated whether initial attack was successful (i.e., the fire was fully contained) using different times of initial engagement, and thus fire size (Moderate = 4 h; High = 2 h; Maximum = 1 h). For the progressive suppression scenario, we assumed ignitions were managed without any initial attack but with subsequent suppression. For fires where initial attack was not successful, we modeled subsequent suppression using suppression functions to relate fireline intensity to the expected proportion of fire suppressed (Fig. 1b). The Maximum suppression filter roughly equates to the maximum possible effectiveness of on-the-ground fire suppression efforts, where suppression becomes virtually impossible above a certain fireline intensity[36].

We calculated daily elliptical distance burned under each suppression scenario as the unsuppressed distance burned (with the assumption that fire actively spread for half of the day length) multiplied by the proportion of fire remaining after suppression. Points on the ellipse were considered permanently extinguished if the daily

distance burned was less than 5 m. Other points on the ellipse could continue burning, and a fire was not considered extinguished until points at all angles were extinguished or until after the 150th day of the fire season. Supplementary Fig. S6 shows the daily fire weather and progression of burning for an example ignition, across all suppression scenarios.

## Fire behavior model inputs

For each ignition, we simulated daily fire weather: windspeed and temporally autocorrelated live and dead fuel moistures. We modeled daily windspeeds as arising from a Weibull distribution, with a single wind direction for each fire event. We simulated canopy fuel moisture as varying temporally across a sinusoidal seasonal trend. We also simulated daily live and dead surface fuel moisture (i.e., 1-, 10-, and 100-h fuel moisture, live woody and herbaceous fuel moisture) using a sinusoidal seasonal trend around a mean seasonal fuel aridity value, with random variation in amplitude and daily, temporally auto-correlated fluctuations. For a given ignition, we assumed a terrain slope of 40% (20.8°) and uniform and continuous fuel loading, meaning that simulated fires never experienced fuel breaks and would continue burning as long as weather conditions allowed for fire spread.

We ran simulations across ranges of mean fuel aridity and fuel loading to emulate changes in both time (i.e., climate change, fuel accumulation) and space (i.e., moving from one climatic region or forest type to another). Fuel aridity and fuel loading ranges spanned 240 years of modeled climate change or fuel accumulation in the Western U.S., respectively. To estimate a rate of yearly fuel aridity change under an RCP 8.5 climate change scenario, we used Ficklin & Novick's[44] projected median increase in summer vapor pressure deficit (VPD) of 0.72 kPa for the continental U.S. from the historical period (1979–2013; midpoint 1996) to the future period (2065–2099; midpoint 2082) to calculate a mean annual increase between midpoint years of 0.00837 kPa yr$^{-1}$. Because fire spread models are input with fuel moisture, we converted VPD to dead fuel moisture, using a statistical relationship parameterized with gridMET data[90] from across Western U.S. temperate conifer forests. To estimate plausible yearly increases in fuel loading, we estimated the slope of modeled dead wood carbon accumulation (i.e., the difference in fuel loads between suppressed and unsuppressed model scenarios) in the Western U.S. from 1980–2010 from Boisramé and colleagues[45]. We assumed that carbon made up 50% of total fuel weight and that 100-h fuels comprised 50% of all dead fuels (in keeping with ratios of Fuel Model 10[91]), yielding a yearly 100-h fuel load increase of 0.036 Mg ha$^{-1}$ yr$^{-1}$.

## Simulation structure

We ran simulations where we varied either mean fuel aridity or fuel loading, while holding the other at constant mean values. A single simulation replicate involved 1000 ignitions, each of which had a unique, randomly simulated ignition day and timeseries of fuel moisture values and windspeeds (i.e., weather scenario). Using this ignition-day and weather scenario, we then simulated fire spread independently for either 1) all levels of fuel loading, while holding mean fire season fuel aridity constant (VPD of 1.17 kPa), or 2) all levels of mean fuel aridity, while holding fuel loading constant (100-h fuel loading of 11.23 Mg ha$^{-1}$). For each of these simulated fire-spread events, we also modeled all four scenarios of fire suppression (Maximum, High, Moderate, and Progressive) in addition to the no-suppression scenario. We then replicated a single simulation 40 times, for a total of 5 million simulated fires across each range of mean fuel aridity or fuel loading (1000 ignitions ×25 levels of fuel aridity or fuel loading ×5 levels of suppression ×40 replicates).

## Summarizing the cumulative impacts of fire suppression

To assess the effect of fire suppression on patterns of fire severity, for each fire we calculated the proportion that burned at high severity (CBI

values ≥ 2.25; sensu Parks & Abatzoglou[53]) and the weighted mean fire severity. For each simulation we calculated the required change in mean fuel aridity or fuel load for fires with no suppression to yield the same fire severity as suppressed fires. We divided this value by estimated yearly rates of change in mean fuel aridity and fuel loading, to evaluate how many years of climate change or fuel accumulation it would take for unsuppressed fires to burn at the same mean fire severity. To investigate how fire suppression affected patterns of burned area, we calculated the total area burned for each fire. For each simulation, we also calculated the average multiplicative yearly rate of increase (△) in burned area across the 240-year ranges of fuel aridity and fuel loading. We calculated the diversity of fire severity for each fire using the method detailed by Steel and colleagues[43], with CBI as the single input fire trait—which is equivalent to the mean absolute deviation of CBI values. We investigated how equally burned area was spread across fires by calculating Lorenz curves of area burned under each suppression scenario.

Within a replicate (i.e., 1000 ignitions × 25 fuel aridity or fuel loading levels), for each suppression scenario we calculated the mean values for parameters of interest, at each level of fuel aridity or fuel loading. We then calculated overall means across the 40 simulation replications and used 95% confidence intervals to display variability. We conducted all simulation and analysis in R[92].

## Data availability

The source data generated in this study have been deposited in a public database[93] [https://doi.org/10.5281/zenodo.10729478].

## Code availability

The R code to reproduce all our results and figures is provided in a public database[93] [https://doi.org/10.5281/zenodo.10729478].

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

## Acknowledgements

We thank Mark Finney and members of the University of Montana Systems Ecology Writing Seminar for suggestions that improved this work. This work was supported by a Graduate Research Fellowship from the National Science Foundation, award 1745048 (M.R.K.). Additional support came from the USDA Forest Service, Rocky Mountain Research Station, Aldo Leopold Wilderness Research Institute through agreement 19-JV-11221639-098 (M.R.K. and A.J.L.) and from the Department of the Interior North Central Climate Adaptation Science Center through Cooperative Agreement G18AC00325 from the United States Geological Survey (P.E.H.). The findings and conclusions in this publication are those of the authors and should not be construed to represent any official USDA, USGS, or U.S. Government determination or policy.

## Author contributions

M.R.K., P.E.H., S.A.P. and A.J.L. designed research; M.R.K. wrote the model and ran computer simulations; M.R.K. analyzed data; M.R.K wrote the paper; M.R.K., P.E.H., S.A.P., W.L.R., N.W. and A.J.L. conducted review and editing.

## Competing interests

The authors declare no competing interests.
