## [Peer Review File · Nature Communications]

Fire suppression makes wildfires more severe and accentuates impacts of climate change and fuel accumulationReviewers' Comments:

Reviewer #1 (Remarks to the Author):

This paper presents a well-designed study to examine the impact of fire suppression bias on the behavior of wildfires across a range of climates and fuel loadings. I especially appreciate the discussion of "socio-ecological impacts," I think the authors made some very important points in this section.

I believe that this manuscript is nearly ready for publication, but have a few relatively minor suggestions for improvement:

Results section: I think it would be helpful to also see some results on the total area burned at high severity as well as total area burned. I appreciate that the rate of increase in area burned is important, but I think land managers and the public are more used to thinking about total area burned. If the authors are concerned this will detract from their main points, they could just mention these results briefly and put a plot in SI for the interested reader.

Line 241: I think this might be more clear if you say something along the lines of "i.e. each modeled fire is completely unaffected by the fires that occurred in prior years, and thus the fire-suppression paradox is not captured in this model."

Line 322: It looks like the value that the authors are using for fuel accumulation from Boisrame and colleagues is from a plot showing the difference in fuel loads between suppressed and unsuppressed model scenarios, rather than a total accumulation rate (which would only be the same if you assume that fires burning unsuppressed result in 0 net accumulation of dead fuels). This may still be a reasonable number to use, as it captures the amount of fuel accumulating due to suppression activities, but the authors should be clear about what this number represents (at least in the supplemental information where there is room to discuss such details). This does not affect the model results of how fire size and severity vary across different fuel loadings, but it might impact some of the interpretation in terms of "years of fuel loading."

Caption for Figure S2: I believe the "high-intensity filter" is supposed to say "high-intensity fire."

The equations given in the "Simulate suppression" section of the SI do not appear to reproduce the curves in Figure S12. Is "exp" not representing the natural exponential equation? Please verify these equations, and either correct them or provide more information so that readers can reproduce the curves in Figure S12.

Reviewer #2 (Remarks to the Author):

The paper reports on modeling the effects of “suppression bias” on wildfire burned area and effects. The justification is in paragraph beginning on line 71. To my knowledge there’s no disagreement that suppression preferentially removes mild and moderate fires (~97% of all ignitions before they grow to 300ac), but this work focuses on the backing portions of large fires (~3% of fires which burn > 90% of the land area). Backing portions have the lowest intensity and spread rate and are easiest to suppress and thus remove from the landscape. It’s interesting that the paper does not address what would appear to be the main effect of suppression – that is, Initial Attack success in removing ~97% of all ignitions, leaving the remaining ~3% to burn under more extreme conditions. Thus, the modeling here targets a much less important factor – and not really captured by the Title or introductory parts of the paper.

While the aim is conceptually defensible, the methodology is considerably more complex than needed to address this problem and to obtain the very general results reported here. The paragraph starting at line 238 presents a long list of caveats associated with the conclusions and as such underscores the need for a very simple, interpretable, and generalized analysis rather than a complex one. The complexity is excessive because 1) the authors acknowledge that they are not making specific predictions for any land area, 2) they employ the simple elliptical fire shapes which have an analytical solution for the distribution of intensities/spread rates within each elliptical footprint (Catchpole et al. 1982, 1992), and 3) that the environmental conditions are highly abstracted from any real geographic location.

To obtain a robust set of results that satisfy the intent and scope of the paper, the authors could have used the intensity/spread rate distributions from Catchpole et al. 1992, and the fraction of backing (below some threshold) to determine the fraction of area removed by suppression. Some of the other factors could also be unambiguously included too, such as different fuel types, different moisture conditions, estimates of fire effects (CBI) etc. The combined results would constitute a geometric argument as to the idealized low intensity proportion of elliptical area burned that is removed by suppression without the need for all of the methodological complications detailed in the supplementary material. For this reason alone, I cannot recommend publication and the revisions to the manuscript would be substantial enough to qualify as a new submission.

To assist possible revisions, I have a list of editorial comments:

1. Title – not descriptive of the actual effort for reasons stated above - not dealing with most of fire suppression on the 97%, just the containment of the 3% that escape.
2. Line 71, 239, 285 – I don’t think the issue is underappreciated by knowledgeable researchers. Low-intensity burning associated with suppression is recognized as an important “progressive” or pro-active management approach. Maybe a better way to say this is poorly quantified.
3. Line 93-95- this is confusing because the rest of the paper is about containment of large fires, not small fires.
4. Line 95-97- these can be easily dealt with using the simplified methods suggested above.
6. Methods (supplemental materials) –There is confusion about what fires are being suppressed. Are you examining containment of small fires under mild conditions or just the 3% that escape? In the former – there are other kinds of attack that are not dealt with there – head vs. tail attack (see Anderson 1989 , Fried and Fried 1996) which change intensity distributions of resulting fire patterns. The effectiveness of

IA for the 97% has been subject to quite a bit of research not cited here. If the fire containment is larger fires only – then much different tactics are employed involving multiple divisions of the perimeter simultaneously. Less research to cite, but still some people have worked on this (Fried et al. 2008, Torn and Fried 1992, Strauss et al. 1989) and a number of others.

Anderson, D.H., 1989. A mathematical model for fire containment. *Canadian Journal of Forest Research*, 19(8), pp.997-1003.

Fried, J.S. and Fried, B.D., 1996. Simulating wildfire containment with realistic tactics. *Forest Science*, 42(3), pp.267-281.

Fried, J.S., Gilles, J.K., Riley, W.J., Moody, T.J., Simon de Blas, C., Hayhoe, K., Moritz, M., Stephens, S. and Torn, M., 2008. Predicting the effect of climate change on wildfire behavior and initial attack success. *Climatic Change*, 87, pp.251-264.

Torn, M.S. and Fried, J.S., 1992. Predicting the impacts of global warming on wildland fire. *Climatic change*, 21(3), pp.257-274.

Strauss, D., Bednar, L. and Mees, R., 1989. Do one percent of the forest fires cause ninety-nine percent of the damage?. *Forest Science*, 35(2), pp.319-328.

Response to reviewer comments

We appreciate the thoughtful and helpful comments from the two anonymous reviewers, which have strengthened and clarified our work. We have fully addressed all reviewer comments in the updated manuscript and offer detailed responses to each comment below. Full references for any cited materials are provided following the comment responses.

Reviewer 1 comments:

Overall: This paper presents a well-designed study to examine the impact of fire suppression bias on the behavior of wildfires across a range of climates and fuel loadings. I especially appreciate the discussion of "socio-ecological impacts," I think the authors made some very important points in this section. I believe that this manuscript is nearly ready for publication but have a few relatively minor suggestions for improvement.

Thank you. We appreciate Reviewer 1's comments and the recognition of the paper's robust design and our discussion of socio-ecological impacts. We have incorporated the suggestions for minor changes and responded in full below.

Results section: I think it would be helpful to also see some results on the total area burned at high severity as well as total area burned. I appreciate that the rate of increase in area burned is important, but I think land managers and the public are more used to thinking about total area burned. If the authors are concerned this will detract from their main points, they could just mention these results briefly and put a plot in SI for the interested reader.

We appreciate this suggestion and have added a figure in the Supplementary Information showing area burned at high severity per fire and included a sentence discussing it in the results. We have not added a figure depicting the total area burned because it effectively already is included. Every simulation has the same number of ignitions (1000), thus the total area burned is identical to the average area burned per fire (i.e., Fig. 3) multiplied by 1000.

Line 241: I think this might be more clear if you say something along the lines of "i.e. each modeled fire is completely unaffected by the fires that occurred in prior years, and thus the fire-suppression paradox is not captured in this model."

We have added text to make it clear that fires are not impacted by fuel conditions changed by other fires.

Line 322: It looks like the value that the authors are using for fuel accumulation from Boisrame and colleagues is from a plot showing the difference in fuel loads between suppressed and unsuppressed model scenarios, rather than a total accumulation rate (which would only be the

same if you assume that fires burning unsuppressed result in 0 net accumulation of dead fuels). This may still be a reasonable number to use, as it captures the amount of fuel accumulating due to suppression activities, but the authors should be clear about what this number represents (at least in the supplemental information where there is room to discuss such details). This does not affect the model results of how fire size and severity vary across different fuel loadings, but it might impact some of the interpretation in terms of "years of fuel loading."

Thank you for clarifying this distinction. We have updated text in the Methods and Supplementary Information to make it clear that this rate is the fuel accumulation due to suppression activities.

Fig. S2 caption: I believe the "high-intensity filter" is supposed to say "high-intensity fire."

We have corrected the caption to "high-intensity fire".

SI: The equations given in the "Simulate suppression" section of the SI do not appear to reproduce the curves in Figure S12. Is "exp" not representing the natural exponential equation? Please verify these equations, and either correct them or provide more information so that readers can reproduce the curves in Figure S12.

Thank you for catching this error. We have corrected the equations in the Supplementary Information.

Reviewer 2 comments:

Overall: The paper reports on modeling the effects of "suppression bias" on wildfire burned area and effects. The justification is in paragraph beginning on line 71. To my knowledge there's no disagreement that suppression preferentially removes mild and moderate fires (~97% of all ignitions before they grow to 300ac).

While we agree that it is widely understood that suppression preferentially removes mild and moderate fires, we know of no studies that quantify the impacts of this "suppression bias" over space and time (i.e., a fire regime). Researchers regularly reference the well-appreciated impact of suppression on increased fuel loads. It is much rarer that papers acknowledge that suppression preferentially removes some types of fire more than others, even though our work shows that this impact of suppression significantly changes resulting patterns of fire behavior at all scales. We thus believe that our work is undeniably novel, timely, and important to researchers, managers, and policymakers alike.

Overall: This work focuses on the backing portions of large fires (~3% of fires which burn > 90% of the land area). Backing portions have the lowest intensity and spread rate and are easiest to suppress and thus remove from the landscape. It's interesting that the paper does not address what would appear to be the main effect of suppression – that is, Initial Attack success in removing ~97% of all ignitions, leaving the remaining ~3% to burn under more extreme conditions. This work focuses on the backing portions of large fires (~3% of fires which burn > 90% of the land area). Backing portions have the lowest intensity and spread rate and are easiest to suppress and thus remove from the landscape. Thus, the modeling here targets a much less important factor – and not really captured by the Title or introductory parts of the paper.

We appreciate the additional comments about the two components of suppression (initial attack & containment of escaped fires). Though initial attack was implicitly captured in the original manuscript (i.e., many fires were successfully contained before they reached 300 acres), in our updated manuscript, we have explicitly added initial attack to the modeling approach. We now incorporate a model simulating initial attack success (Hirsch et al. 1998), in which initial attack success declines as fire intensity and fire size increase. We have also included an additional figure showing the proportion of fires successfully contained before 300 acres. This update has not fundamentally changed the main findings in the original manuscript; however, it captures suppression with increased fidelity, and we are grateful for this suggested improvement. We have updated our language throughout to make it explicitly clear that we are modeling both initial attack and containment of escaped fires.

Overall: While the aim is conceptually defensible, the methodology is considerably more complex than needed to address this problem and to obtain the very general results reported here. The paragraph starting at line 238 presents a long list of caveats associated with the conclusions and as such underscores the need for a very simple, interpretable, and generalized analysis rather than a complex one. The complexity is excessive because 1) the authors acknowledge that they are not making specific predictions for any land area, 2) they employ the simple elliptical fire shapes which have an analytical solution for the distribution of intensities/spread rates within each elliptical footprint (Catchpole et al. 1982, 1992), and 3) that the environmental conditions are highly abstracted from any real geographic location.

To obtain a robust set of results that satisfy the intent and scope of the paper, the authors could have used the intensity/spread rate distributions from Catchpole et al. 1992, and the fraction of backing (below some threshold) to determine the fraction of area removed by suppression. Some of the other factors could also be unambiguously included too, such as different fuel types, different moisture conditions, estimates of fire effects (CBI) etc. The combined results would constitute a geometric argument as to the idealized low intensity proportion of elliptical area burned that is removed by suppression without the need for all of the methodological complications detailed in the supplementary material. For this reason alone, I cannot recommend publication and the revisions to the manuscript would be substantial enough to qualify as a new submission.

Thank you, we appreciate these comments. Reviewer 2 supports our fundamental conceptual aim but argues that our analysis is overly complex. They raise two main points: 1) that our results could be achieved through simplified methods (*sensu* Catchpole 1992), and 2) that we do not make predictions for a specific landscape (i.e., abstracted from a specific geographic location).

To the first point, we agree that methods ought to be no more complex than necessary. The aim of our study requires the simplest modeling framework that: 1) is robust to changing conditions (i.e., fire weather), 2) can represent the critically important transition from surface to crown fire, and 3) can incorporate long-term directional change in fuel loads and climate. We carefully investigated the method suggested by Reviewer #2 (*sensu* Catchpole et al. 1992), including meeting with a globally renowned expert in fire modeling, to discuss the modeling approach. However, the “analytical solution” approach suggested by Reviewer 2 does not meet the fundamental goals of this study, for two reasons. First, the elliptical intensity distribution method that Reviewer 2 suggests does not extend to crown fire, where some points on the fire perimeter expand much faster, and the perimeter becomes non-elliptical. Second, and more critically, the suggested methods are incompatible with changing fire conditions (e.g., wind speed) inherent in modeling fire across time. As a recent fire modeling textbook puts it, “The fireline intensity patterns and the proportions shown [by Catchpole et al. 1992] will be greatly altered if a fire experiences a change in wind speed.... The resulting distributions for these conditions must be generated by simulation models.” (“Wildland Fire Behaviour”, Finney et al, 2021; p. 39; emphasis added).

We did implement the recommendation of Reviewer #2 to use the fireline intensity calculation method of Catchpole and colleagues (1982) and we thank the reviewer for that suggestion. We believe that our current approach strikes a thoughtful balance between simplicity and important nuance. We also point out that the “methodological complications” in the Supplementary Information are no more complex than off-the-shelf fire spread models like FARSITE or BehavePlus. Our methods only have the appearance of complexity because we chose to document our protocols and parameters thoroughly instead of merely referencing a stock fire spread model.

To the second point raised by Reviewer #2, we acknowledge that all models are tradeoffs between fine-scale precision and large-scale generalizability (Levins 1966). We deliberately chose not to link our predictions to a single, small-scale land area to reduce the possibility that readers would interpret our results as applying *only* to a specific location. Rather, we use a wide range of hypothetical landscapes (i.e., a wide range of fuel loading and climate) to incorporate the variability and uncertainty inherent in real landscapes and show that our results are applicable globally—in any landscape where suppression preferentially removes some types of fire more than others. While this inherently comes at the cost of some precision, our modeling approach is well suited to the goals of our paper and follows a long history of such modeling approaches (e.g., Levins 1966; Franklin & Forman 1987). We have added a section in the Methods clarifying why we chose this approach and how it is generalizable yet realistic.

Title: Not descriptive of the actual effort for reasons stated above - not dealing with most of fire suppression on the 97%, just the containment of the 3% that escape.

We maintain that the title accurately and descriptively captures the main findings shown by our data—that suppression (both initial attack and containment of escaped fires) makes wildfires more severe and heightens the impacts from climate change and fuel accumulation. As noted above, we have updated our methodology and language to make it explicitly clear that we are modeling both initial attack and containment of escaped fires.

Lines 71, 239, 285: I don't think the issue is underappreciated by knowledgeable researchers. Low-intensity burning associated with suppression is recognized as an important "progressive" or pro-active management approach. Maybe a better way to say this is poorly quantified.

We agree that the fact that suppression preferentially removes mild and moderate fires is known by researchers and managers. However, we argue that the consequences of this fact (i.e., the emerging patterns that our results reveal) are under-appreciated. There has been no research to quantify the consequences of the suppression bias at the scale of a fire regime or in the context of directional change in climate or fuel loading. Without evidence of prior research that explores these consequences, we believe "under-appreciated" is an accurate term.

We have incorporated the suggestion of "poorly quantified" in places where we are referring to the fact of the suppression bias but have retained "under-appreciated" when referring to the consequences that emerge given this reality.

Lines 93–95: This is confusing because the rest of the paper is about containment of large fires, not small fires.

We have updated our modeling approach to include a model simulating initial attack success. The Introduction, Methods, and Results now make it explicit that we are modeling both initial attack and containment of escaped fires.

Lines 95–97: These can be easily dealt with using the simplified methods suggested above.

The method suggested by Reviewer 2 (Catchpole et al. 1992) does not extend to the non-elliptical perimeters caused by crown fires and is not applicable to conditions that vary across time (e.g., changing wind speeds). We have thus retained our approach of using a simulation (see above comments for additional response).

SI Methods: There is confusion about what fires are being suppressed. Are you examining containment of small fires under mild conditions or just the 3% that escape? In the former – there are other kinds of attack that are not dealt with there – head vs. tail attack (see Anderson 1989, Fried and Fried 1996) which change intensity distributions of resulting fire patterns. The effectiveness of IA for the 97% has been subject to quite a bit of research not cited here. If the fire containment is larger fires only – then much different tactics are employed involving multiple divisions of the perimeter simultaneously. Less research to cite, but still some people have worked on this (Fried et al. 2008, Torn and Fried 1992, Strauss et al. 1989) and a number of others.

We have updated our methodology and language to make it clear that we are modeling both initial attack and containment of escaped fires. We appreciate the suggestion of additional relevant references and have incorporated them and the ideas from them in many places throughout the updated manuscript.

References:

- Catchpole, E. A., Mestre, N. D., & Gill, A. (1982). Intensity of fire at its perimeter. *Australian Forest Research*. <https://www.frames.gov/catalog/26098>
- Catchpole, E. A., Alexander, M. E., & Gill, A. M. (1992). Elliptical-fire perimeter- and area-intensity distributions. *Canadian Journal of Forest Research*, 22(7), 968–972. <https://doi.org/10.1139/x92-129>
- Finney, M. A., McAllister, S. S., Grumstrup, T. P., & Forthofer, J. M. (2021). *Wildland Fire Behaviour*. CSIRO Publishing.
- Franklin, J. F., & Forman, R. T. T. (1987). Creating landscape patterns by forest cutting: Ecological consequences and principles. *Landscape Ecology*, 1(1), 5–18. <https://doi.org/10.1007/BF02275261>
- Hirsch, K. G., Corey, P. N., & Martell, D. L. (1998). Using Expert Judgment to Model Initial Attack Fire Crew Effectiveness. *Forest Science*, 44(4), 539–549. <https://doi.org/10.1093/forestscience/44.4.539>
- Levins, R. (1966). The Strategy of Model Building in Population Biology. *American Scientist*, 54(4), 421–431.

REVIEWERS' COMMENTS

Reviewer #1 (Remarks to the Author):

This revision has adequately addressed my concerns from the original manuscript. I believe this version is acceptable for publication.

Reviewer #3 (Remarks to the Author):

Review of NCOMMS-23-26430A-Z

Nice read. well written. Only have a few comments that should be addressed. The main one, which would require an addition of a paragraph or two regards the effect of fire load on the landscape on the suppression strategies. I recommend acceptance of the manuscript

The title should reflect that the study is a modelling one, not based on empirical data. This is a model exercise, that as described by the authors is constrained by the “trade-offs between fine scale precision and large scale generalizability”. How realistic is the approach is debatable. It is a interesting modelling framework but the results are certainly largely influenced by the underlying assumptions.

An important aspect that was left out of the discussion is the concept of fire load that is not addressed in the suppression scenarios. If one lets fires burn in the landscape under mild to moderate burning conditions, when a period of higher fire potential arises, there is no chance to suppress or contain all the fires freely spreading on the landscape. This will result on the fact that instead of a few large fires associated with new ignition during the bad fire weather event, one would be facing possibly tens of fires spread with high intensity on the landscape. This in fact is the scenario of past large fire events throughout the world (the three million acres burned by the NW US 1910 great fires was the result of hundreds of fires in the landscape; similar situation with the 1939 Australian black Friday fires). A paragraph or two on this very real aspect should be included.

Some detailed comments

The authors use the term “realistic” 5 times, as if to give weight to their approach. But most of the assumptions are quire general, so “realistic” is a strong word for what is in the text.

This simulation is only relative to forest areas? Or also grassland areas?

What is the assumed return interval to an area burned? I could not find it.

Line 140 – not sure what the author means by “realistic” fuel loading values. I am not sure they are realistic at all. I suggest changing to “assumed fuel loading values”.

Line 191: We show through a modelling framework how decades....

Line 241 – Do not use words such as novel. Keep it simple and let the science do the talking. I could not

see any novel management strategies. In this section it seems the authors are just repeating results.

Line 323: 100h fuels? The moisture of extinction is related to the fine fuels, the 1hr time lag fuels in the US system. 100 h fuels do not vary daily. Even if one considers a weekly time step, the variation in 100 h fuels is low unless there is rain. Or is this assumption to ensure that fires only stop when the moisture contents of large fuel are high? Which seems sound.

Response to review comments

We appreciate the second round of helpful comments from the two anonymous reviewers, which have further improved the manuscript. We have fully addressed all reviewer comments in the updated manuscript and offer detailed responses to each comment below.

Reviewer #1 comments:

This revision has adequately addressed my concerns from the original manuscript. I believe this version is acceptable for publication.

Thank you. We appreciate these comments.

Reviewer #3 comments:

Review of NCOMMS-23-26430A-Z

Nice read. well written. Only have a few comments that should be addressed. The main one, which would require an addition of a paragraph or two regards the effect of fire load on the landscape on the suppression strategies. I recommend acceptance of the manuscript

Thank you. We have addressed the comment regarding fire load below.

The title should reflect that the study is a modelling one, not based on empirical data. This is a model exercise, that as described by the authors is constrained by the “trade-offs between fine scale precision and large scale generalizability”. How realistic is the approach is debatable. It is a interesting modelling framework but the results are certainly largely influenced by the underlying assumptions.

We have retained the original title, as we feel that it summarizes the main findings of the paper in a concise and accurate manner. As noted above in our in-depth response to this comment (above), we clearly describe that this study uses a simulation approach, in the abstract and throughout the manuscript.

An important aspect that was left out of the discussion is the concept of fire load that is not addressed in the suppression scenarios. If one lets fires burn in the landscape under mild to moderate burning conditions, when a period of higher fire potential arises, there is no chance to suppress or contain all the fires freely spreading on the landscape. This will result on the fact that instead of a few large fires associated with new ignition during the bad fire weather event, one would be facing possibly tens of fires spread with high intensity on the landscape. This in fact is the scenario of past large fire events throughout the world (the three million acres burned

by the NW US 1910 great fires was the result of hundreds of fires in the landscape; similar situation with the 1939 Australian black Friday fires). A paragraph or two on this very real aspect should be included.

Thank you for this comment. We have added a paragraph to the Discussion to discuss this important aspect. While the model may underestimate the suppressed size of fires, because the model assumes nearly complete in-effectiveness of suppression during extreme weather events, it already partially incorporates the resource scarcity that would arise from a large “blow-up” of many fires.

The authors use the term “realistic” 5 times, as if to give weight to their approach. But most of the assumptions are quite general, so “realistic” is a strong word for what is in the text.

We have removed the use of the term “realistic” throughout, and instead used “plausible” when referring to the fact that model parameters and inputs are informed by actual ranges of values.

This simulation is only relative to forest areas? Or also grassland areas?

We state in the methods and discussion that our models are based on fuels and climate of forested areas, however, by using a wide range of fuel loading, we demonstrate that the impacts of the fire suppression bias are relevant to any location worldwide (i.e., forest, grassland, savannah, etc.) where suppression is used. Any time lower-intensity fire is more easily removed than higher-intensity fire, this will lead to a “suppression bias”, regardless of the ecosystem type.

What is the assumed return interval to an area burned? I could not find it.

Our model does not incorporate fire return intervals. We explicitly state that our model simulates fires independently, and thus does not incorporate fuel feedbacks (e.g., the fuel limitation or the effects of the fire suppression paradox). Instead, our model isolates the impacts of the fire suppression bias alone.

Line 140 – not sure what the author means by “realistic” fuel loading values. I am not sure they are realistic at all. I suggest changing to “assumed fuel loading values”.

We removed the term “realistic” throughout. In this particular instance, we changed the language to “across the simulated range of fuel loading values”.

Line 191: We show through a modelling framework how decades....

We updated this text as suggested to indicate that we used a modelling framework.

Line 241 – Do not use words such as novel. Keep it simple and let the science do the talking. I could not see any novel management strategies. In this section it seems the authors are just repeating results.

We have removed all instances of “novel” and similar words from the manuscript. We have also edited this paragraph to avoid repeating results.

Line 323: 100h fuels? The moisture of extinction is related to the fine fuels, the 1hr time lag fuels in the US system. 100 h fuels do not vary daily. Even if one considers a weekly time step, the variation in 100 h fuels is low unless there is rain. Or is this assumption to ensure that fires only stop when the moisture contents of large fuel are high? Which seems sound.

Thank you for catching this error. We are using the R function “rothermel” from the package “firebehaviorR” to generate fire behavior modeling, and you are correct in pointing out that the moisture of extinction parameter is not related to 100-hour fuels. We have corrected this in the methods text to reflect what the R script is calculating.